# Short-Term Forecast of Indoor CO_2_ Using Attention-Based LSTM: A Use Case of a Hospital in Greece

**DOI:** 10.3390/s25175382

**Published:** 2025-09-01

**Authors:** Christos Mountzouris, Grigorios Protopsaltis, John Gialelis

**Affiliations:** Department of Electrical and Computer Engineering, University of Patras, 265 04 Patras, Greece; mountzou@ece.upatras.gr (C.M.); g.protopsaltis@ac.upatras.gr (G.P.)

**Keywords:** indoor air quality, time-series forecasting, CO_2_, LSTM, machine learning

## Abstract

Given the significant implications of indoor air pollution for physical and mental health, well-being and productivity, indoor air quality (IAQ) is of critical importance. CO_2_ is a prevalent indoor air contaminant and represents a key determinant for IAQ characterization. This study collected sensed air pollution and climatic data from a hospital environment in Greece and employed Long Short-Term Memory (LSTM) neural network variants with progressively increased architectural complexity to predict indoor CO_2_ concentration across future horizons ranging from 15 min up to 180 min. Among the examined variants, the attention-based LSTM exhibited the most consistent performance across the forecasting horizons. Incorporating additional predictors reflecting climatic conditions, air pollution and occupancy status within the hospital settings, the multivariate attention-based LSTM further enhanced its predictive performance with an MAE of 8.9 ppm, 16.7 ppm, 31.2 ppm, 38.9 and 39.5 ppm for 15 min, 30 min, 60 min, 120 min, and 180 min ahead, respectively.

## 1. Introduction

With modern people spending up to 90% of their time in enclosed spaces, indoor air quality (IAQ) represents a critical concern for public health, with exposure to poor IAQ linked to a substantial health burden. Each year, the WHO reports over 3.2 million premature deaths and the Global Burden of Disease estimates more than 8.09 million deaths worldwide attributable to poor IAQ [1]. Poor IAQ also has adverse effects on well-being and productivity [2,3]. Recent research evidence has revealed that indoor air pollution consistently exceeds outdoor levels by 2 to 10 times, and occasionally by up to 100 times [4,5].

Carbon dioxide (CO_2_) is a prevalent indoor air pollutant and a critical determinant for characterizing IAQ. Human respiration constitutes a main indoor CO_2_ emissions source, while significant contributors also include combustion and fuel-burning processes such as heating with solid or fossil fuels, cooking, and smoking [6]. Due to insufficient ventilation in indoor spaces, CO_2_ emissions infiltrating from outdoors further deteriorate IAQ, with fossil fuel combustion in transportation being the main CO_2_ emissions driver in urban areas [7].

The indoor air is contaminated by several substances and compounds, with the most critical being carbon oxides (COx), particulate matter (PM), nitrogen oxides (NOx), ozone (O_3_), volatile organic compounds (VOCs), radon, and toxic metals [8]. Health organizations and regulatory bodies have established thresholds and exposure durations for pollutants based on their health implications. The American Society of Heating, Refrigerating and Air-Conditioning Engineers (ASHRAE) recommends that indoor CO_2_ levels should remain below 1000 ppm, with indoor–outdoor differential not exceeding 700 ppm [9]. This threshold is also reflected in 10 more guidelines, while 800 ppm and 1500 ppm frequently appear as upper boundaries in regional regulations [10]. Under typical conditions, indoor CO_2_ levels range between 400 and 1000 ppm.

Recent advancements in Artificial Intelligence (AI), Machine Learning (ML), big data analytics, and sensors have enabled proactive and informed IAQ management strategies to support healthier indoor spaces and promote energy efficiency [11]. Heating, ventilation, and air-conditioning (HVAC) systems are integral parts of IAQ management; however, their operation is associated with increased energy demands [12]. Leveraging predictive analytics, pattern recognition, and anomaly detection in IAQ, actionable insights are produced for HVAC optimization with the dual objective to minimize energy costs and maintain a healthy indoor environment [13,14].

The persistent need for accurate IAQ predictions motivated this study, which set out to evaluate Long Short-Term Memory (LSTM) neural networks concerning indoor CO_2_ concentrations over short temporal horizons, spanning from 15 min up to 3 h. The rationale behind this architectural selection was grounded in their strong capacity to learn intricate temporal patterns. This study implemented both univariate and multivariate LSTMs to investigate both temporal dependencies inherent in CO_2_ and interconnected temporal associations between CO_2_, climatic conditions, air pollution and occupancy.

A literature review was conducted on prior research studies that employed ML models for short-term CO_2_ forecasting indoors. Deep learning (DL) approaches were also identified, allowing a fairer comparison with the LSTM models developed in the present study. Predictive performance was reported using two standard regression error metrics: Mean Absolute Error (MAE) and Root Mean Squared Error (RMSE).

J. Kallio et al. [15] evaluated ML and DL models in short-term CO_2_ forecasting within office settings using a one-year data collection. Decision Tree and Random Forest regression models maintained an optimal balance between accuracy and computational efficiency for CO_2_ predictions up to 15 min ahead, utilizing a 10-min lookback window. Human presence was identified as the most influential predictor. Multi-Layer Perceptrons (MLPs) did not outperform the tree-based models despite their capacity to capture more intricate dependencies. Under a similar setup in office settings, N. Kapoor et al. [16] compared several ML models for instant CO_2_ forecasts, including linear, decision tree, support vector machines, MLP, and Gaussian process regression (GPR). Among the models examined, optimized GPR achieved the lowest MAE of 3.35 ppm with an RMSE of 4.20 ppm. Apart from the linear model, all the fine-tuned model variants produced reasonably well forecasts, underscoring the ML potential in CO_2_ forecasting. Y. Zhu et al. [17] proposed a low-cost, IoT-enabled IAQ monitoring system that leveraged LSTM models to predict CO_2_ levels 1 min ahead. Uni- and bi-directional LSTM architectures with single and stacked layers were developed, using 10-min lookback windows. The bi-directional LSTM attained the lowest MAE of 8.95 ppm with an RMSE of 16.77 ppm.

K.U. Ahn et al. [18] introduced a novel hybrid approach that coupled white-box, physics-based models with black-box, data-driven modeling to produce short-term CO_2_ forecasts. A Bayesian Neural Network (BNN) was utilized to address the uncertainty of the mass balance equation model by estimating the model’s unknown or hard-to-measure input variables, such as aggregated ventilation rates, infiltration, and interzonal airflow. The study examined forecasting horizons of 5 min, 10 min, and 15 min ahead using a 27-day data collection from residential settings. The results were quite promising, with the hybrid model’s deterioration over increased forecasting horizons being less pronounced, achieving a Mean Bias Errors (MBE) of −0.01% and −1.28% for the living room and the bedroom, respectively. P. Garcia-Pinilla et al. [19] conducted extensive experiments across 15 schools in Spain, comparing the predictive gains in CO_2_ forecasting from statistical, ML and DL models, examining future horizons ranging from 10 min up to 4 h. For horizons up to 10 min, ML and DL models exhibited marginal gains compared to simpler statistical models, while DL models demonstrated robust predictive performance with increasing forecasting horizons. Temporal Convolutional Networks (TCN) achieved the lowest MAE of 29.19 ppm for near-term horizons and N-HiTS achieved the lowest MAE of 36.72 ppm for longer ones. S. Taheri and A. Razban [20] implemented ML and DL models to predict indoor CO_2_ concentrations in university classrooms for 1 h, 6 h and 24 h ahead. MLP achieved the lowest MAE of 30.16 ppm and 55.19 ppm for 1 h and 24 h ahead, respectively.

Aiming to optimize ventilation systems, E. Athanasakis et al. [21] employed LSTM and Convolutional Neural Network (CNN) models to predict indoor CO_2_ concentrations 5 min ahead, with a dilated CNN resulting in the lowest MAE of 4.56 ppm. To enable fast deployment in real-world applications, G. Segala et al. [22] implemented a low-complexity CNN using air temperature, relative humidity, and CO_2_ as predictors. This CNN achieved an RMSE around 15 ppm at 15 min future horizon, highlighting that even low complexity DL approaches could deliver substantial performance gains.

The remainder of this study is articulated as follows. Section 3 describes the core methods and materials employed in this study. Section 4 details the LSTM variant evaluation.

## 2. Methods and Materials

Figure 1 illustrates the high-level methodological framework employed in this study to forecast indoor CO_2_ concentrations at future horizons of 15 min, 30 min, 60 min, 120 min, and 180 min. The time-series data collected from the four (4) sensing devices installed within hospital environment were preprocessed, imputing missing values, resampling with 1-min resolution, and aggregating into single measurements per time step using median. Subsequently, a feature engineering was performed to compute linear correlations, autocorrelations inherent in CO_2_, and mutual information, assessing their predictive relevance of available features. Using the most informative features, input sequences were constructed for univariate and multivariate LSTM models, resampled to a 5-min resolution, and split into training and testing sets with 80:20 ratio. Using equal-length lookback windows and forecasting horizons, predictions for each horizon were generated with multiple LSTM variants, including simple, deep, attention-based, and dropout-regularized architectures.

### 2.1. Data Collection and Description

For the present study, a dataset comprising air quality and climatic time-series measurements was sourced from the indoor settings of the General Hospital Thriassio (GHT) located at Elefsina, Attica, Greece, and specifically, from the Secretary of Emergency Services zone. This zone serves as the primary hospital’s one-stop shop for the incoming patients and visitors to emergency services, and as a result, it experiences considerable and persistent occupation throughout the working days. Elevated occupancy patterns in this zone are also observed during the on-call duty days of GHT, although they do not follow any consistent, recurring temporal pattern. To gather data, a small-scale network of four (4) wall-mounted sensor units was installed in this hospital zone, each mounted on one of the four walls delineating the space, and was built using the open-source, modular SmartCitizen Kit (SCK) 2.1 platform (Fab Lab Barcelona, Institute for Advanced Architecture of Catalonia, Barcelona, Spain).

The monitoring period comprised two distinct one-month intervals from 21 September 2024 to 21 October 2024, and from 22 January 2025 to 22 February 2025, with measurements recorded at 1-min resolution. The dataset corresponding to the first monthly interval, denoted as $D_{1}$, comprised a total of 43,629 time-series measurements for each sensor unit, while the dataset corresponding to the second interval, denoted as $D_{2}$, comprised a total of 40,109 measurements. Across both datasets, time-series measurements were collected for air pollutants of CO_2_, PM_2.5_, and TVOC, for climatic conditions of air temperature, relative humidity, and barometric pressure, and for noise and luminosity levels, which served as indirect occupancy indicators for this zone (Table 1). These variables, denoted as $x_{t,\;*}\in R^{8}$, were treated as the candidate predictors for the target variable of CO_2_, denoted as $y_{t}=F(x_{t,\;*})$, where $F:R^{8}\to R$ at each specific time step t.

### 2.2. Data Preparation and Preprocessing

The data preparation and preprocessing phase commenced with the exploration of $D_{1}$ and $D_{2}$ for missing values, as the sequential models require continuous and valid measurements. Missing values probably resulted from invalid payload transmissions, payload decoding errors, and occasional sensor malfunctions, since the data collection process was not interrupted at any time. Given the 1-min time interval between successive measurements, the backward imputation strategy was employed as a robust approach to replace the missing values in the dataset. This simple, yet effective strategy for measurements with low sampling resolution propagates the next known to replace the missing one. A total of 129 missing values were imputed for variables representing predictors $\left(x_{t,\;*}\right)$, as was also input for 12 missing values associated with the target variable $\left(y_{t}\right)$ of CO_2_ concentrations.

Under the assumption that the indoor space experienced uniform environmental conditions, a sensor fusion process was performed to aggregate the time-series measurements collected from the four sensor units into a single, representative measurement for each 1-min time interval. The median was used as the aggregation function to ensure that each single measurement is not affected by transient disturbances, sensor-specific anomalies, and local perturbations.

### 2.3. Dataset Exploration

The data exploration phase involved computing descriptive statistics for the dataset’s variables across the two monitoring periods. Central tendency and dispersion measures were calculated to provide a detailed presentation of their principal distributional characteristics and patterns.

As presented in Figure 2, the CO_2_ concentration distributions were broadly comparable and consistent across the two monitoring periods, where $D_{1}$ and $D_{2}$ exhibited means of 516.2 ppm and 520.3 ppm, and standard deviations of 122.7 ppm and 120.4 ppm, respectively. These considerable variations indicated pronounced CO_2_ fluctuations around a nearly identical average in the two monitoring periods. Both medians were lower by 7% compared to the corresponding means, indicating slightly right-skewed CO_2_ distributions. Further examining distributional characteristics, the 75th percentiles in $D_{1}$ and $D_{2}$ were nearly identical at 565 ppm and 566 ppm, respectively, while the 95th percentile was higher in $D_{2}$ by 3.45%, reflecting a marginal prevalence of elevated CO_2_ concentrations during the latter period. On the whole, the underlying distributional characteristics in the CO_2_ concentration across the two monitoring periods substantiate the assumption of stable prevailing conditions in the examined hospital settings.

The air temperature distributions demonstrated notable differences across the two monitoring periods, as presented in Figure 3. Both datasets showed a close arrangement between their central tendency measures with a consistent variance of 1.4 °C. In particular, $D_{1}$ showed a mean of 23.2 °C with a higher median of 0.1 °C, whereas $D_{2}$ exhibited 26.3 °C for both measures. The pronounced differences in central tendency were probably attributable to the thermal regulation of indoor environment through HVAC systems, and the relatively low variance in temperatures indicated stable thermal conditions. For both datasets, the IQRs were also similar, ranging from 22.4 °C to 24.1 °C in $D_{1}$, and from 25.1 °C to 27.3 °C in $D_{2}$.

The relative humidity distributions exhibited substantial variation between the two monitoring intervals, as presented in Figure 4, reflecting the interplay between outdoor climate, building usage, and indoor air regulation. During the first monthly interval $\left(D_{1}\right)$, the mean and median relative humidity were 56.0% and 56.5%, respectively, with the IQR ranging from 49.9% to 61.9%. This indicated a generally humid indoor environment, with values frequently clustering in the upper half of the observed range. In stark contrast, the second monthly interval $\left(D_{2}\right)$ showed a marked decrease in relative humidity, with the mean dropping to 29.95%, the median to 32.0%, and the IQR shifting down between 24.0% and 35.0%, pointing to a much drier indoor environment during the second interval. This reduction in both central tendency and spread may be attributed to the operation of heating systems during the winter months, which commonly leads to reduced indoor relative humidity.

The PM_2.5_ concentration distribution showed a significant reduction in the second monitoring interval, accompanied by a marked variance reduction, as illustrated in Figure 5. During the first interval, the mean was 10.9 ug/m^3^ with a high standard deviation of 11.9 ug/m^3^ and an IQR ranging between 0.0 ug/m^3^ and 16.0 ug/m^3^. In contrast, the mean decreased by 53.3% during the second interval with a lower standard deviation of 6.5 ug/m^3^ and an IQR ranging between 0.0 ug/m^3^ and 7.0 ug/m^3^. The zero PM_2.5_ concentration recorded during these two monitoring periods indicated excellent IAQ, with concentrations frequently below the sensors’ detection thresholds.

Similarly to PM_2.5_, TVOC concentrations also demonstrated a noticeable decrease in central tendency and variability measures in the second interval, as presented in Figure 6. During the first interval, TVOC had a mean of 185.9 ppb with a standard deviation of 145.3 ppb and an IQR ranging from 89.0 ppb to 257.0 ppb, reflecting elevated indoor TVOC levels with increased variance. During the second intervals, the TVOC mean dropped to 147.42 ppb, representing an approximate 20.7% reduction, while the standard deviation decreased to 85.33 ppb, corresponding to a reduction of 41.3%. In this interval, the IQR was narrower and between 81.0 ppb and 193.0 ppb, indicating an overall IAQ improvement and a substantial reduction in TVOC fluctuations.

The distribution of barometric pressure did not result in any interesting findings, remaining remarkably stable across the two monitoring intervals. These small variations are attributable to normal atmospheric fluctuations rather than any meaningful shift. The mean values were 101.1 kPa and 101.7 kPa for the first and second intervals, with low corresponding variabilities of 0.57 kPa and 0.54 kPa.

### 2.4. Feature Engineering

#### 2.4.1. Exploring Linear and Monotonic Relationships with CO_2_

Adhering to the principle of parsimony, Pearson’s (r) and Spearman’s $\left(\rho \right)$ correlation coefficients were computed to explore linear and monotonic relationships between the candidate predictors and the CO_2_ concentration. Figure 7 displays the results from Pearson’s correlation and Figure 8 the results from Spearman’s correlation. Following the established interpretation guidelines for Pearson’s correlation, no candidate predictor exceeded even the lower threshold of $\left|r\right|\geq 0.4$ to denote a moderate linear relationship with CO_2_ concentration. The indirect occupancy indicators emerged as the most influential predictors, with noise levels exhibiting the maximum correlation levels with r=0.376 and  ρ=0.419, and the light levels following with r=0.246 and  ρ=0.339. The consistently higher Spearman’s coefficients for indirect occupancy indicators suggested the presence of non-linear monotonic relationships with CO_2_ concentration, indicating that the occupancy effects may follow threshold or saturation patterns rather than simple linear increases. Among the air pollutants, TVOC (r=0.227, ρ=0.266) showed a weak to moderate linear correlation with CO_2_ concentration, consistent with its mutual dependence on human occupancy patterns, while PM2.5 (r=0.135, ρ=0.282) resulted to a weak linear relationship with CO_2_ concentration. Climatic conditions demonstrated negligible linear correlations, with barometric pressure, air temperature, and relative humidity resulting in $\left|r\right|\leq 0.1$. These initial findings on Pearson’s and Spearman’s correlation magnitudes underscored that CO_2_ concentration is probably governed by more complex, non-linear interactions with candidate predictors across the occupancy indicators, climatic conditions, and air pollutants.

#### 2.4.2. Autocorrelation of CO_2_

While Pearson’s (r) correlation coefficient can adequately capture linear dependencies between candidate predictors and CO_2_ concentration, the inclusion of lagged CO_2_ concentrations itself could also reveal strong linear associations, thereby uncovering autocorrelation within specific lookback windows. To reduce the high-frequency noise of the 1-min resolution in CO_2_ measurements, a 5-min aggregation using the median was applied, resulting in 24 lags spanning a 120-min temporal window. The autocorrelation function (ACF) exhibited statistically significant values for all these 24 lags $\left(\left|ACF\right|>0.02,\;\;p<0.05\right)$, following a smooth, monotonic decay pattern throughout this temporal window without any oscillatory behaviors. An ACF value of 0.99 at lag 1 indicated near-perfect linear correlation between current CO_2_ concentrations and those measured 5 min earlier, demonstrating remarkable short-term persistence. At 30 min, the ACF value remained considerably high at over 0.85, while at 60 min it decreased to 0.69, still maintaining substantial autocorrelation patterns. Even at lag 24, the ACF value sustained a meaningful level of 0.44, confirming that CO_2_ retains forecasting utility over 120-min periods. The Partial ACF (PACF) revealed that CO_2_ exhibits primary dependence on lag 1 and secondary negative correlation at lag 2, with PACF values of 0.99 and −0.38, respectively, while contributions beyond lag 2 remained negligible. The observed negative PACF at lag 2 aligns with the gradually decaying ACF, supporting the presence of short-term persistence in CO_2_ concentrations, followed by a mild correction effect at longer lags, which is a pattern characteristic of autoregressive processes with direct dependencies primarily at lags 1 and 2.

#### 2.4.3. Granger Causality Analysis

After examining whether the current values of candidate predictors exhibit linear correlation with CO_2_ concentration, as well as the autocorrelation of CO_2_ itself, the Granger causality test was employed to determine whether lagged values of the candidate predictors could enhance the predictability of future CO_2_ concentrations. Notably, predictors related to indoor climatic conditions were found to significantly enhance the prediction of CO_2_ concentrations, notwithstanding their previously limited linear associations. Noise level, air temperature, relative humidity, and PM_2.5_ exhibited highly significant Granger causality (p<0.001) across all the 24 time lags, indicating sustained predictive strength for CO_2_ concentration throughout the entire 120-min temporal window. Barometric pressure also shows highly significant Granger causality between lag 4 and lag 11 (p<0.001), moderate significance between lag 12 and lag 16 (p<0.01), and marginal significance from lag 17 onwards (p<0.05). This diminishing significance at higher lags indicated that the predictive relationship of barometric pressure weakens as the temporal gap increases. Light level demonstrated marginal predictive capacity (p<0.05) for up to 40 min with an optimal forecasting performance at lag 4 (p=0.0098), which suggested a localized and time-sensitive relationship with CO_2_ concentrations. Lastly, TVOC displayed significant Granger causality exclusively at lag 1 (p=0.0035), indicating that its predictive effect on CO_2_ concentration is contemporaneous and immediate, rather than delayed.

#### 2.4.4. Mutual Information (MI)

To explore complex, non-linear dependencies between the candidate predictors and CO_2_ concentrations, the mutual information (MI) measure was utilized. As a measure of association, MI captures the total shared information between variables, providing sensitivity to both linear and non-linear dependencies. All the candidate predictors related to climatic conditions resulted in relatively higher MI scores than those related to occupancy status and air pollutants, despite exhibiting weaker linear correlations with CO_2_ concentration. Based on the MI measure, barometric pressure emerged as the most informative predictor (MI=0.526), with its negligible linear correlation (r=0.031) with CO_2_ highlighting the presence of a pronounced non-linear dependency. Relative humidity and air temperature showed substantial shared information content with CO_2,_ resulting in MI=0.459 and MI=0.353, respectively, which also contradicted their weak linear correlations and highlighted the presence of non-linear dependencies. Notably, noise level exhibited a relatively low MI=0.115 in contrast to its stronger linear correlation with CO_2_ (r=0.376), indicating a predominant linear relationship. Light levels (MI=0.225) and TVOC (MI=0.276) maintained similar strengths of association under both linear and non-linear evaluation information content, whereas PM_2.5_ provided a moderate amount of additional non-linear information (MI=0.201).

#### 2.4.5. Polynomial and Interaction Terms

Recognizing the potential for synergistic effects among candidate predictors, polynomial and interaction terms were generated to evaluate underlying dependencies with CO_2_ concentrations that may not be represented by the original predictors alone. All the second-degree polynomial terms demonstrated MI scores nearly close to those of their corresponding linear terms; MI=0.5208 for the second-degree barometric pressure, MI=0.4571 for the second-degree relative humidity, and MI=0.3568 for the second-degree air temperature. Therefore, incorporating higher-degree terms did not offer any significant predictive advantage. Interaction terms revealed significant synergistic relationships between candidate climatic predictors. The product of barometric pressure and relative humidity achieved the highest interaction score (MI=0.4528); however, it was closely approximating that of relative humidity alone. Notably, air temperature interactions with barometric pressure (MI=0.4100) and relative humidity (MI=0.3809) yielded higher mutual information scores than air temperature individually, indicating that incorporating these interactions enhances predictive capacity relative to using air temperature alone. From an aggregated perspective, the MI achieved by the original candidate predictors was marginally superior in terms of both maximum $\left({MI}_{max}=0.5223\right)$ and average values $\left({MI}_{avg}=0.381\right)$ compared to their polynomial counterparts $\left({MI}_{max}=0.5208,{MI}_{avg}=0.3789\right)$. Therefore, despite the lower individual peak $\left({MI}_{max}=0.4528\right)$, interaction terms demonstrated the most consistent performance with ${MI}_{avg}=0.3393$ and ${MI}_{std}=0.0585$.

### 2.5. Modeling

#### 2.5.1. Naïve Baseline

In the present study, the Last Observation Carried Forward (LOCF) method was used to establish a naïve baseline against which to evaluate the implemented LSTM variants. LOCF is a non-parametric, persistence method, which propagates the latest observed measurement across the desired future time horizon. Under this zero-change assumption, LOCF serves as the minimum error benchmark for more complex and sophisticated approaches in time-series forecasting, particularly for time-series exhibiting low temporal variance and irregular fluctuation patterns that deviate from smooth monotonic trends. Typically, the CO_2_ concentration in indoor spaces demonstrates such erratic temporal behavior, particularly in this study, where human occupancy serves as the main source of CO_2_ emissions, without interference from other significant sources. Let $y_{t,CO2}$ representing the concentration of CO_2_ at a specific time step t, then the LOCF method can be mathematically described by Equation (1), where H denotes the forecasting horizon.(1)$${\hat{y}}_{t+H,CO2}=y_{t,CO2}\;\forall \;H\in N^{+}$$

#### 2.5.2. LSTM Architectures

As already mentioned, this study implemented four (4) LSTM architectures with progressive structural complexity. The rationale behind this incremental approach is twofold: to identify the LSTM architecture with optimal predictive performance, while at the same time accounting for the computational overhead introduced by the specific architecture, acknowledging that increased complexity often entails diminishing marginal gains that may not justify the associated overhead. For each architecture, both univariate and multivariate LSTM modeling paradigms were implemented, thus exploring the predictive capacity of both autoregressive CO_2_ dependencies and exogenous variables in short-term CO_2_ concentration forecasting, in accordance with the feature engineering insights. The LSTM models produce single-step predictions for 15 min, 30 min, 60 min, 120 min, and 180 min ahead, using equal-length lookback windows. This proportional architectural design for the lookback windows’ length ensured alignment between the temporal context of the input sequences and output, thereby maintaining a balanced retention of relevant historical information. Given that CO_2_ concentration demonstrates stable dynamics with minimal variability over successive 1-min periods, the input sequences employed 5-min sampling intervals to attenuate noise artifacts.

To fine-tune the hyperparameters for each LSTM architecture, a grid search with 3-fold cross-validation was performed, resulting in the configurations described in Equations (2)–(7). Given the increased demands for computational resources for this optimization process, the present study did not conduct an exhaustive grid search. The search space encompassed the 8, 16, 32, and 64 units per layer and dropout rates between 0.1 and 0.4 with increments of 0.1, subject to the constraint that dropout rates between consecutive layers must differ by 0.1. During the hyperparameter tuning, two batch sizes of 32 and 64 samples were considered, along with learning rates of $\eta =1\times 10^{-3}$ and $\eta =1\times 10^{-2}$. To ensure effective model convergence and mitigate the risk of overfitting, an early stopping rule of 3 epochs was incorporated for monitoring the validation loss.

The first LSTM model $f_{1}(\cdot )$ implemented a three-layer sequential architecture: an LSTM layer with 64 hidden units configured to process the entire input sequence and return only the final hidden state, a dense intermediate layer with 16 neurons using ReLU activation function for non-linear feature transformation, and a single-neuron linear output layer for direct CO_2_ concentration regression. The training of this shallow model used batches of 32 input sequences with a learning rate of $\eta =1\times 10^{-2}$, resulting in relatively rapid convergence within 9 epochs. By employing conservative architectural complexity, this model established an additional baseline, complementary to the naïve one, which demonstrated the minimal computational overhead for capturing temporal dependencies in CO_2_ concentration using DL. The batch size resulting from the optimization was 32 samples.

Building upon the previous model, the potential of deeper LSTM architectures was explored. Thus, the second LSTM model $f_{2}(\cdot )$ stacked three layers with a progressively reduced dimensionality of 64, 32, and 16 units. The cascading structure allowed the first two LSTM components to retain sequential outputs for continued temporal modeling, whereas the terminal layer compressed the entire sequence into a unified representation. This hierarchical temporal processing was complemented by an 8-neuron dense layer with ReLU activation for feature refinement, terminating in a single-neuron output layer for CO_2_ concentration forecasting. Utilizing batch length sizes of 32 and learning rate of $\eta =1\times 10^{-3}$, the training process for this model reached convergence at epoch 11. Relative to the first LSTM approach, this model demonstrated considerably higher complexity with increased depth and parameter count, thus enabling more nuanced temporal pattern learning at the expense of computational resources.

Attention mechanisms constitute a popular approach for dynamic weighting in RNNs, allowing for selective focus on historically relevant periods with patterns or significant external events, rather than being constrained by the limited information capacity of the final hidden state. To overcome this limitation, the third LSTM model $f_{3}(\cdot )$ was augmented with such mechanism, employing an architecture consisting of a 64-unit LSTM layer that processed the entire input sequence, while maintaining time step-level outputs, followed by a custom attention layer that computes temporal relevance scores for each time step based on learned query vectors. The attention mechanism applied softmax normalization to generate attention weights that sum to unity and produce a context-aware representation by computing the weighted sum of LSTM hidden states, thereby enabling the model to dynamically focus on historical periods with similar CO_2_ concentration patterns. The architecture concluded with a 16-neuron dense layer with ReLU activation for additional feature transformation and a single-neuron output layer for CO_2_ concentration forecasting. Regarding the training process, this model used batch sizes of 64 and learning rate $\eta =1\times 10^{-3}$, with slow yet stable convergence at epoch 16.

Dropout regularization is one of the most common techniques used in RNNs to prevent over-reliance on specific temporal patterns by zeroing out a small fraction of neuron outputs during training. In this sense, this technique was applied in the fourth LSTM model $f_{4}(\cdot )$, incorporating three stacked LSTM layers with 64, 32, and 16 hidden units, respectively. A progressively increasing dropout rate of 0.1, 0.2, and 0.3 was applied, adhering to the results from the hyperparameter configuration process. The architecture was completed with a 16-neuron dense layer succeeded by 0.4 dropout—the highest rate applied to mitigate the substantial overfitting risk inherent in fully connected layers—and concluded with a single-neuron output layer for CO_2_ concentration forecasting. With batch sizes of 64, this model achieved the slowest convergence at 22 epochs.

For clarity and to facilitate methodological validation, the mathematical formulations for the LSTM models are presented. Let $X\in R^{T\times d}$ be the input sequence with length T in d-dimensional feature space and $\hat{y}\in R$ be the predicted CO_2_ concentration at future horizon $h\in N^{+}$. Equations (2) and (3) describe the formulation for the first $f_{1}(X)$ and the second $f_{2}(X)$ LSTM models, respectively, where σ represents the ReLU activation function and $b_{i}$ represents the corresponding bias term.(2)$$f_{1}(X)=W_{2}\cdot \sigma \left(W_{1}\cdot {LSTM}_{64}(X)+b_{1}\right)+b_{2}$$(3)$$f_{2}(X)=W_{2}\cdot \sigma \left(W_{1}\cdot {LSTM}_{16}\left({LSTM}_{32}\left({LSTM}_{64}\left(X\right)\right)\right)+b_{1}\right)+b_{2}$$

The third LSTM model $f_{3}(X)$ is mathematically formulated through Equations (4) and (5), where $\alpha_{i}$ represent the attention weights and $h_{i}$ represent the hidden states of the 32-unit LSTM layer.(4)$$f_{3}(X)=W_{2}\cdot \sigma \left(W_{1}\cdot Att\left({LSTM}_{64}(X)\right)+b_{1}\right)+b_{2}$$(5)$$Att(H)=\sum_{i}\alpha_{i}h_{i},\mathrm{where}\;\alpha_{i}=\mathrm{softmax}\left(tanh\left(W_{a}h_{i}\right)\right)$$

The fourth LSTM model $f_{4}(X)$ is mathematically formulated through Equations (6) and (7), where ${Drop}_{k}$ represents a dropout rate of k.(6)$$f_{4}(X)=W_{2}\cdot {Drop}_{0.4}\left(\sigma \left(W_{1}\cdot \psi (Χ)+b_{1}\right)\right)+b_{2}$$(7)$$\psi (Χ)={Drop}_{0.3}\cdot \left({LSTM}_{16}\left({Drop}_{0.2}\left({LSTM}_{32}\left({Drop}_{0.1}\left({LSTM}_{64}\right)\right)\right)\right)\right)$$

### 2.6. Performance Evaluation

To evaluate the performance, two standard error metrics used in regression problems were employed: the Mean Absolute Error (MAE), which averages the absolute differences between predicted $\left(\hat{y_{i}}\right)$ and actual values $\left(y_{i}\right)$ regardless of the error direction, and the Root Mean Squared Error (RMSE), which poses a quadratic term ${\left(y_{i}-\hat{y_{i}}\right)}^{2}$ to penalize significant deviations from the actual values $\left(y_{i}\right)$.(8)$$MAE=\frac{1}{n}\sum_{i=1}^{n}\left|y_{i}-\hat{y_{i}}\right|$$(9)$$RMSE=\sqrt{\sum_{i=1}^{n}\frac{{\left(y_{i}-\hat{y_{i}}\right)}^{2}}{n}}$$

## 3. Results and Discussion

### 3.1. Preparation of Training and Validation Input Sequences

Dataset $D_{1}$ was allocated as the training set for model fitting, whereas dataset $D_{2}$ functioned as the holdout test set to provide an unbiased estimate of the predictive performance in independent, previously unseen data. The walk-forward validation technique was applied across the dataset $D_{1}$ using 5 folds to address the limitations of single chronological train-validation splits, thus ensuring robust generalization, prevention of data leakage, and facilitating multiple independent evaluations of a model’s performance across different time periods. In each fold, a split ratio of 80/20 was used, with a fixed training window of 8000 samples and a validation window of 2000 samples, where the training and validation periods progressed chronologically through the dataset $D_{1}$.

Let ${\hat{y}}_{t+h,\;u}$ denote a single-step prediction for the CO_2_ concentration produced by a univariate LSTM model at a specific future horizon $h\in \left\{15,30,\;60,\;120,\;180\right\}$. These models use a lookback window length w and a sampling rate s=5, such that $s\times \left(w-1\right)=h$. Each univariate model also receives an input sequence with a series of past CO_2_ measurements, represented by the input vector $X_{u}$, as in Equation (10).(10)$$X_{u}=\left[y_{t-s(w-1)},\;y_{t-s(w-2)},\;..,\;y_{t-s},\;y_{t}\right]$$

Likewise, let ${\hat{y}}_{t+h,\;m}$ denote a single-step prediction for the CO_2_ concentration produced by a multivariate LSTM model at a specific future horizon $h\in \left\{15,30,\;60,\;120,\;180\right\}$. These models use a lookback window length w and a sampling rate s=5, such that $s\times \left(w-1\right)=h$. The multivariate LSTM models incorporate past measurements of climatic conditions, air pollutants, and occupancy indicators, besides the past CO_2_ concentration. Specifically, these models receive input sequences of equal length to the horizon h for relative humidity $\left(x_{t,\;rh}\right)$, noise levels $\left(x_{t,\;nl}\right)$, PM_2.5_ $\left(x_{t,\;pm}\right)$, and the interaction term of air temperature and relative humidity $\left(x_{t,\;tr}\right)$. When applicable, the input sequence incorporated barometric pressure $\left(x_{t,\;bp}\right)$ consisting of lagged values in 5-min intervals from 60 min up to 20 min ago, and for light $\left(x_{t,\;lm}\right)$ levels included values from 20 min prior to t up to present. The input vector $X_{m}$ for multivariate LSTM models is described by Equation (11), with Equations (12)–(14) presenting the corresponding input sequences for individual predictors.(11)$$X_{m}=\left[y_{win},\;x_{win,\;tr},\;x_{win,\;rh},x_{win,\;nl},x_{win,\;pm},x_{win,\;rb},\;x_{[0:20],\;lm},x_{[20:60],\;bp}\right]$$
(12)$$x_{win,\;*}=\left[x_{t-s(w+1)},\;x_{t-s(w+2)},\;..,\;x_{t-s},\;y_{t}\right]$$(13)$$x_{[0:\alpha ],\;lm}=\left[x_{t-\alpha ,lm},\;x_{t-\alpha +s,\;lm},\;..,\;x_{t,lm}\right],\;where\;\alpha =40$$(14)$$x_{[b:c],\;bp}=\left[x_{t-c,bp},\;x_{t-c+s,\;bp},\;..,\;x_{t-b,np}\right],\;where\;b=40\;and\;c=60$$

### 3.2. Naïve Baseline—Performance Evaluation

Given that indoor CO_2_ concentration is primarily influenced by human occupancy and exhibits gradual temporal dynamics, the naïve baseline served as a minimum performance threshold to evaluate the necessity of advanced modeling strategies. This persistence-based benchmark was especially relevant for near-term prediction horizons such as 15 min and 30 min ahead, where the assumption that the current CO_2_ level would hold steady is empirically supported due to the inherent inertia in indoor air quality dynamics. At longer-term horizons, this zero-change assumption functioned as an agnostic reference that highlights the limitations of static predictions and clarifies the necessity for dynamic modeling.

As presented in Figure 9, the naïve baseline demonstrated a degradation pattern with increasing future horizons, confirming the pronounced inverse relationship between predictive accuracy and lead time in naïve approaches. At the 15-min interval, it achieved relatively modest error rates, and specifically an MAE of 30.2±5.8 ppm and an RMSE of 47.2±8.8 ppm. As the forecast horizon extended, the predictive performance deteriorated substantially, with an increased MAE of around 53% at the 30-min interval relative to the 15-min one. The 60-min ahead forecasts showed further degradation, nearly double compared to those obtained at the 15-min interval, with an MAE of 64.4±8.5 ppm and an RMSE of 96.7±13.1 ppm. The deterioration pattern became most pronounced for longer horizons, with 120-min and 180-min forecasts reaching 89.8±12.4 ppm and 105.0±18.7 MAE, respectively. From these results, it became evident that the prediction’s uncertainty exhibited an exponential growth. Importantly, the increasing standard deviations across longer forecasting horizons, from ±5.8 ppm for 15-min to ±18.7 ppm for 180 min ahead, indicated not only declining average performance but also reduced reliability of the persistence assumption.

### 3.3. Performance Evaluation for the Univariate LSTM Models

Building upon the minimum performance benchmarks established by the naïve baseline, univariate LSTM architectures were evaluated. Although feature selection and prior research works [15,18,20,21,22] highlighted that incorporating additional predictors related to climatic conditions, air pollutants and occupancy status could enhance the performance of the short-term CO_2_ forecasting, this univariate approach offered fundamental insights into the autoregressive properties and temporal dependencies inherent in CO_2_. To facilitate model referencing, the following nomenclature was employed: the architecture specified in Equation (2) is hereafter referred to as simple LSTM, Equation (3) as deep LSTM, Equation (4) as attention-based LSTM, and Equation (6) as LSTM with dropout.

As presented in Figure 10, the simple LSTM model achieved a substantial forecasting advantage against the naïve baseline across all the temporal horizons. At 15-min and 30-min intervals, it achieved an MAE of 12.1±2.5 ppm and 21.9±3.9 ppm, respectively, translated into relative improvements of 59.9% and 52.4%. Across longer forecasting horizons, the simple LSTM also exhibited superior performance relative to the naïve baseline, reducing the MAE by 45.8%, 50.2%, and 56.3% for 60 min, 120 min and 180 min ahead. Although similar error reduction rates were also reflected in the RMSE, indicative of effective attenuation in larger or less predictable CO_2_ fluctuations, the higher variance highlighted considerable uncertainty.

As presented in Figure 11, the deep LSTM achieved further MAE reductions of 8.9%, 1.5%, 7.7%, and 7.1% for the 15-min, 60-min, 120-min, and 180-min ahead forecasts, whereas for the 30-min ahead forecast, it yielded an MAE deterioration of 5.7%. Compared to the simple univariate LSTM model, the lower accuracy at the 30-min interval indicated that the model’s increased complexity may have introduced overfitting or vanishing gradient, which offset the benefits of greater depth. Comparable trends were also observed for RMSE, reinforcing that the deeper LSTM not only reduces typical forecasting errors but also suppresses larger, infrequent deviations more effectively.

The univariate attention-based LSTM led to substantial gains in predictive accuracy across the mid-range and long-range forecasting horizons compared to the simple and the deep model variants, with the corresponding errors presented in Figure 12. For the 30-min and 60-min forecasting horizons, where the deep architecture struggled, the attention-based LSTM reduced the MAE by 25% and 21%, respectively, and the RMSE by 21% and 19%. These gains underscored that the augmentation of LSTM with attention layers enabled the dynamic weighing of relevant time steps within the input sequence, thus enhancing the model’s ability to capture more nuanced temporal dependencies. For 120 min and 180 min ahead, the LSTM with the attention mechanism also resulted in more pronounced results compared to the previously presented models, with 11.9% and 14.2% additional improvement in MAE. At shorter horizons, the deep and attention-based LSTM models demonstrated an almost identical MAE and RMSE, which indicated that the temporal relationships up to 15 min ahead could be adequately captured by recurrent layers alone.

Despite the incorporation of regularization into the univariate LSTM architecture, which also introduced a significant computational overhead, the variant with dropout did not deliver any performance gains compared to the attention-augmented model, as illustrated in Figure 13. The results revealed that this model is mostly comparable to the deep LSTM variant, achieving relative MAE and RMSE reductions of around 20% for 30 min and 60 min ahead. For the extended future horizons of 120 min and 180 min, LSTM with dropout continued to provide moderate MAE improvements of around 12%. In contrast, the MAE at 15 min ahead was higher by 18.8% and the RMSE by 7.5%, underscoring that stochasticity introduced by the dropout may hinder the model’s ability to fully leverage the available short-term temporal structure.

### 3.4. Performance Evaluation for the Multivariate LSTM Models

The multivariate simple LSTM model did not outperform its univariate counterpart, with near-identical MAEs across all the forecasting horizons, as presented in Figure 14. At the 15-min interval, both approaches achieved an MAE of around 12.0 ppm, with negligible improvements of 0.2 ppm and 1.2 ppm to be observed in the MAE for 120-min and 180-min ahead forecasts. A relatively modest MAE reduction was attained for the multivariate simple LSTM at 30 min and 60 min ahead, translated into MAE reductions of 4.57% and 5.16%, respectively. These narrow performance gaps indicated that the multivariate simple LSTM model failed to deliver better predictive outcomes, and also underscored that such a simple LSTM architecture could not capture intricate temporal patterns between CO_2_ concentration and exogenous variables.

On the contrary, with the multivariate deep LSTM model, it became evident that incorporating a higher-dimensional space of relevant predictors could lead to enhanced accuracy. The predictive outcomes of this modeling approach are presented in Figure 15. Although it recorded higher errors at the longer horizons of 120 min and 180 min ahead, with an MAE of 39.6±10.1 ppm and 40.5±10.8 ppm, respectively, the multivariate model achieved a notable MAE reduction of 35.1% and 14.2% for the corresponding 30-min and 60-min intervals. When benchmarked against the multivariate simple LSTM, this variant showed a consistent and marked improvement across all the forecasting horizons.

A sustained superior predictive performance relative to its univariate counterpart was revealed for the attention-based LSTM, maintaining a predictive advantage across all the evaluated forecasting horizons, as shown in Figure 16. For 15 min ahead, it achieved an MAE of 8.9±1.3 ppm translated into an MAE reduction of 7.3% compared to the univariate attention-based model, while for 30 min ahead, it further reduced the error by 9.2% with an MAE of 16.7±2.5 ppm. A decreased MAE of 6.5% was observed for both the 60-min and 12-min ahead forecasts, and for 180 min ahead, it achieved an MAE of 39.5±9.9 that indicated an improved performance of 3.9%. The persistent advantage of the attention-based multivariate LSTM underscored that the inclusion of auxiliary predictors, when processed through attention layers, provides complementary information that cannot be fully exploited by univariate or conventional recurrent models alone.

Much like the previous results, the multivariate LSTM with dropout also achieved persistently lower errors than its univariate counterpart across the examined horizons as illustrated in Figure 17, with notable relative MAE reductions of 22.8% and 14.1% for 15 min and 30 min ahead. At these two intervals, MAEs were marginally higher by 0.1 ppm and 0.3 ppm compared to the multivariate attention-based LSTM. At the 60-min interval, it resulted in the lowest performance gains with an MAE reduction of 3% relative to its univariate counterpart, while for 120 min and 180 min ahead, the MAE decreased by 6.8% and 6.1%, respectively. Notwithstanding the substantial relative performance gains, these MAEs were slightly higher than those attained from attention-augmented multivariate LSTM. Considering the associated computational burden with regularization, the attention mechanism stands out as the most efficient and practical approach for short-term forecasting of CO_2_ concentration in real-world applications under the given context.

## 4. Conclusions

This study evaluated univariate and multivariate LSTM models adopting progressively increased architectural complexity for indoor CO_2_ forecasting across horizons spanning from 15 min up to 180 min ahead, leveraging a two-month data collection from the Secretary of Emergency Services of the GHT.

The ACF analysis revealed that CO_2_ exhibited strong temporal persistence up to 30 min ago and retained moderate autocorrelation up to 1 h ago. The Granger causality testing indicated statistically significant temporal dependencies between CO_2_ and additional predictors, with the most salient ones observed for noise level, air temperature, relative humidity and PM_2.5_ over a 120-min temporal window. Considerable shared information was identified between CO_2_ and air temperature, barometric pressure, and relative humidity, underscoring performance gains from incorporating these climatic conditions into the input sequences. The indirect occupancy indicators of light and noise levels demonstrated the highest Pearson’s correlation coefficients despite their relatively low MI scores, probably indicating an underlying linear relationship with CO_2_.

All the univariate LSTM models outperformed the naïve baseline across the examined forecasting horizons within the studied environment. A shallow univariate LSTM architecture achieved an MAE reduction of 59.9% and 52.4% for 15 min and 30 min ahead, while attention-based and regularized univariate LSTM architectures resulted in further error reduction. Considering the computational overhead associated with dropout, the attention-based LSTM maintained the optimal balance between computational efficiency and predictive accuracy, establishing itself as the evaluation benchmark against multivariate approaches. For 15 min and 30 min ahead, the univariate attention-based LSTM attained an MAE of 9.6 ppm and 18.4 ppm, while for 60 min ahead, the MAE was 26.7 ppm. For 120 min and 180 min ahead, it yielded near identical MAEs of 33.3 ppm and 32.9 ppm.

Among the examined configurations and within the experiment’s context, the most substantial predictive gains in CO_2_ forecasting were achieved with a multivariate attention-based LSTM architecture. This architecture’s relative superior performance reflected the multifaceted interplay among climatic conditions, air pollution, and occupancy governing indoor CO_2_, and underscored the attention mechanism’s capabilities to prioritize the most informative time steps. For 15 min, 30 min and 60 min ahead, this attention-based architecture achieved an MAE of 8.9 ppm, 16.7 ppm, and 33.4 ppm, respectively. Even for longer forecasting horizons, the MAE remained below 40 ppm, with comparable MAE of 38.9 ppm and 39.5 ppm for 120 min and 180 min ahead.

Compared to the predictive outcomes from related research works, the proposed multivariate attention-based LSTM attained competitive or superior performance over most forecasting horizons within the studied hospital environment. It should be stressed that each study explored datasets with diverse contextual, seasonal, and building characteristics, and incorporated different predictors, including CO_2_, climatic conditions, and occupancy patterns; therefore, generalizability beyond the studied environment remains to be demonstrated.

At 15 min ahead, the proposed multivariate attention-based LSTM achieved an MAE of 8.9 ppm, comparable to the MAE of 8.95 ppm that was reported by Y. Zhu et al. [17] using a bidirectional LSTM at a considerably shorter 1-min horizon. By contrast, E. Athanasakis et al. [21] reported a lower MAE of 4.56 ppm using a dilated CNN, examining also a considerably shorter 5-min horizon. At 30 min ahead, the proposed multivariate attention-based LSTM achieved an MAE of 16.7 ppm, outperforming the TCN of García-Pinilla et al. [19] that attained an MAE of 29.19 ppm. At 60 min ahead, the MAE was 33.4 ppm, which is comparable to the 30.16 ppm achieved by the MLP of Taheri and Razban [20]. Even for longer horizons of 120 min and 180 min, the proposed multivariate attention-based LSTM maintained an MAE below 40 ppm, yielding accuracy on par with the best multi-hour results reported by García-Pinilla et al. [19], and substantially outperformed the 24-h ahead forecasts with an MAE of 55.19 ppm in Taheri and Razban [20].

## 5. Future Work

This study demonstrates the practical value of attention-based LSTM models for short-term CO_2_ forecasting in indoor environments. By integrating climatic conditions, air pollutants, and occupancy status, the proposed approach yields accurate and reliable predictions that go beyond simple monitoring. The model supports energy-efficient operation of portable embedded devices, allowing them to adjust sensing schedules dynamically, reduce redundant measurements, and extend battery life. It also enables proactive IAQ management: instead of relying on reactive ventilation, which is limited by slow system response times and mixing dynamics, predictive insights allow timely interventions before CO_2_ thresholds are exceeded. This ensures healthier indoor conditions, reduces occupant exposure, and prevents the energy-intensive boosts often required by late corrective actions. Such predictive control is valuable across diverse settings—including offices, classrooms, healthcare facilities, and homes—where differences in room size, occupancy density, and HVAC responsiveness can otherwise result in prolonged periods of elevated CO_2_. By anticipating concentrations 15–180 min in advance, the proposed model provides a universal safeguard, simultaneously enhancing comfort, compliance, and energy efficiency.

Three main directions warrant further future investigation. First, this study relied on a single-site dataset with a specific sensor configuration and monitoring window, which may introduce instrumentation and temporal-coverage biases; external validation on datasets collected from diverse buildings, climates, and occupancy regimes is necessary to assess generalization. As a second direction, future research should prioritize the direct incorporation of occupancy, given its substantial influence on CO_2_ dynamics. The reliance on noise and luminosity levels to infer occupancy status represents a methodological limitation, and addressing this constraint could yield significant improvement in predictive performance. The third direction should be the integration of predictive models into digital solutions like digital twins and virtual sensing to confirm their practicality in real-world applications. As a fourth direction, future research studies should focus on identifying the dominant physical and environmental factors driving variations in CO_2_, PM_2.5_, and VOC concentrations, in order to complement the forecasting models with a stronger causal understanding and support more effective mitigation strategies.

## Figures and Tables

**Figure 1 sensors-25-05382-f001:**
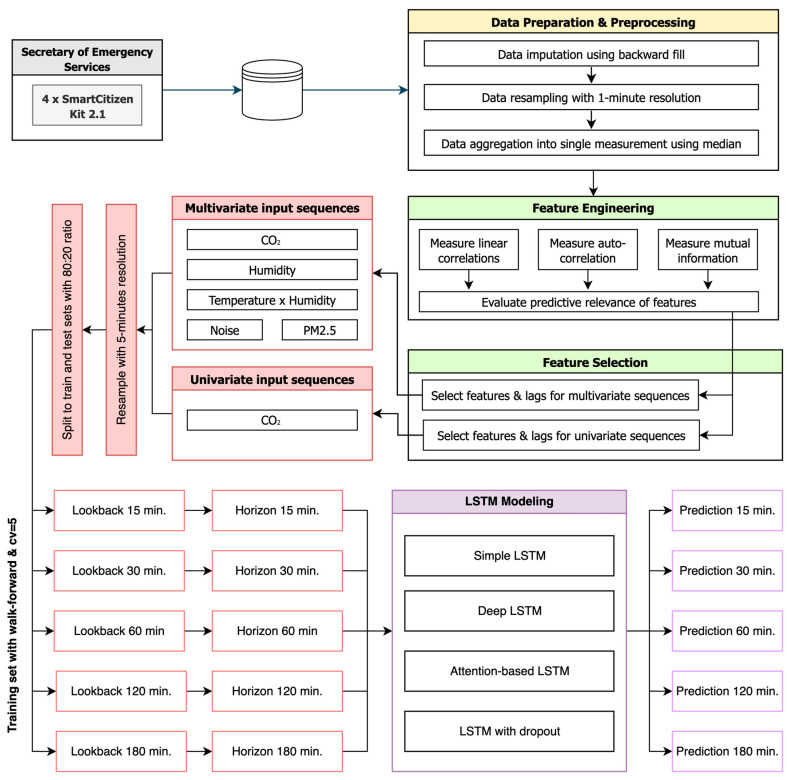
The proposed methodological framework for indoor CO_2_ forecasting.

**Figure 2 sensors-25-05382-f002:**
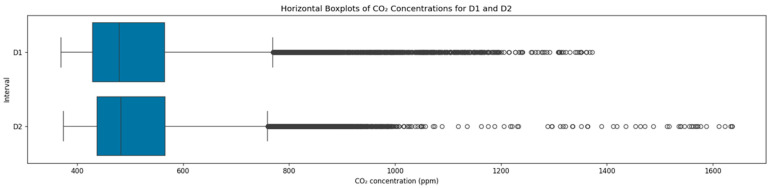
Distribution of CO_2_ concentrations across the two datasets.

**Figure 3 sensors-25-05382-f003:**
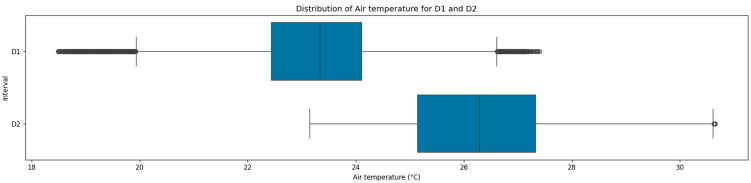
Distribution of air temperatures across the two datasets.

**Figure 4 sensors-25-05382-f004:**
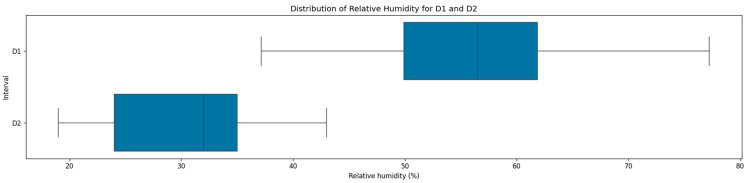
Distribution of relative humidity across the two datasets.

**Figure 5 sensors-25-05382-f005:**
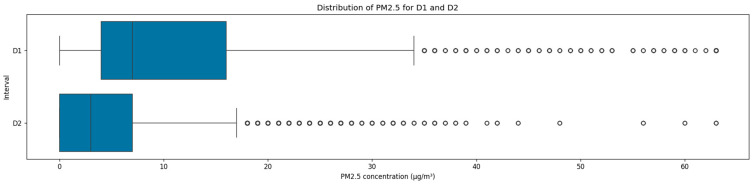
Distribution of PM_2.5_ concentrations across the two datasets.

**Figure 6 sensors-25-05382-f006:**
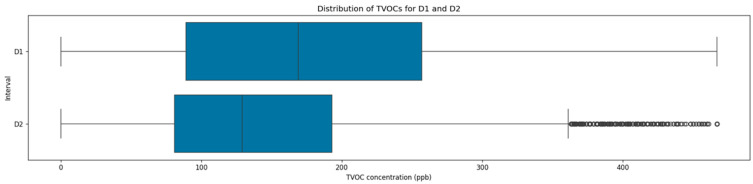
Distribution of TVOC concentrations across the two datasets.

**Figure 7 sensors-25-05382-f007:**
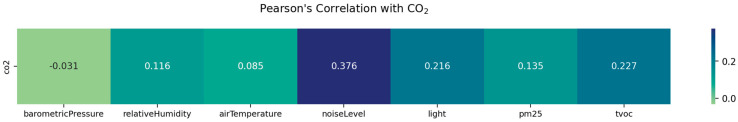
Pearson’s correlation between candidate predictors and CO_2_.

**Figure 8 sensors-25-05382-f008:**
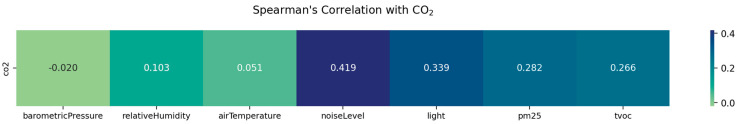
Spearman’s correlation between candidate predictors and CO_2_.

**Figure 9 sensors-25-05382-f009:**
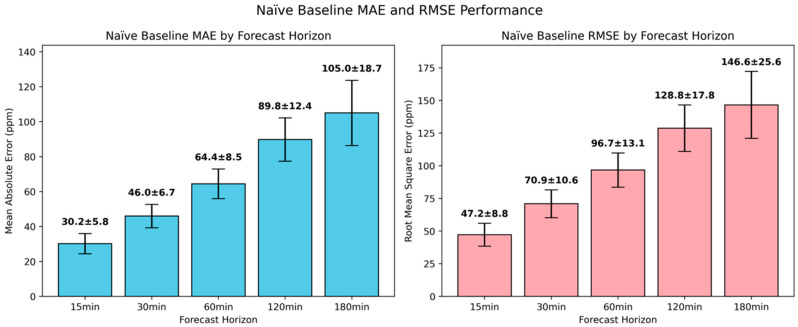
Evaluation of naïve baseline in short-term CO_2_ forecasting using MAE and RMSE.

**Figure 10 sensors-25-05382-f010:**
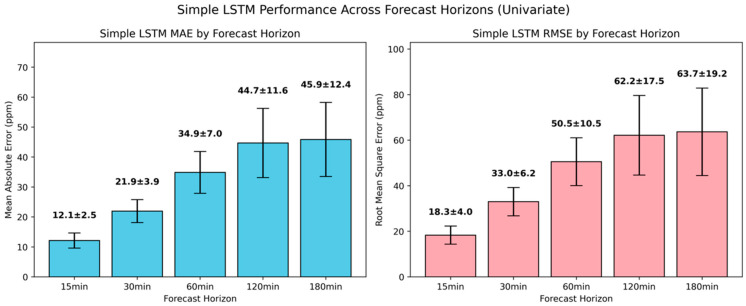
Evaluation of univariate simple LSTM model in short-term CO_2_ forecasting using MAE and RMSE.

**Figure 11 sensors-25-05382-f011:**
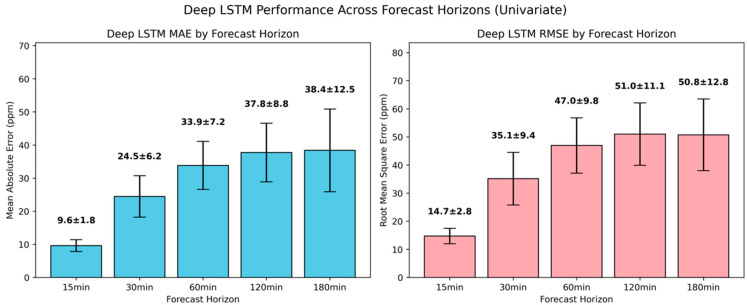
Evaluation of univariate deep LSTM model in short-term CO_2_ forecasting using MAE and RMSE.

**Figure 12 sensors-25-05382-f012:**
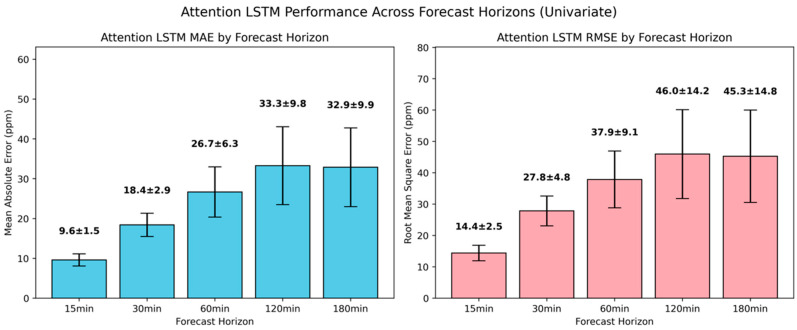
Evaluation of univariate attention-based LSTM model in short-term CO_2_ forecasting using MAE and RMSE.

**Figure 13 sensors-25-05382-f013:**
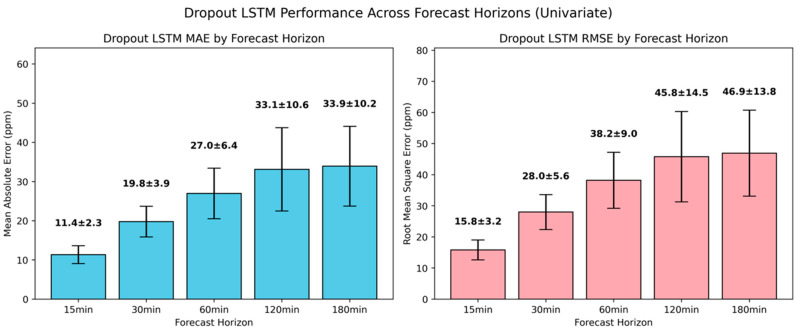
Evaluation of univariate LSTM with dropout model in short-term CO_2_ forecasting using MAE and RMSE.

**Figure 14 sensors-25-05382-f014:**
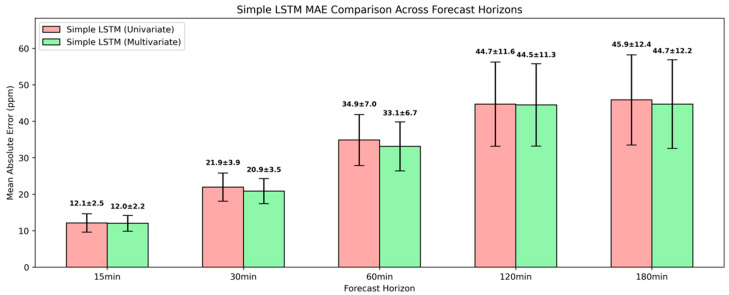
Evaluation of multivariate simple LSTM model in short-term CO_2_ forecasting using MAE and RMSE.

**Figure 15 sensors-25-05382-f015:**
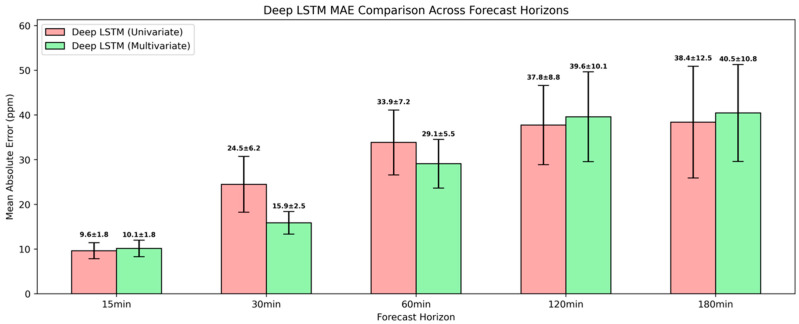
Evaluation of multivariate deep LSTM model in short-term CO_2_ forecasting using MAE and RMSE.

**Figure 16 sensors-25-05382-f016:**
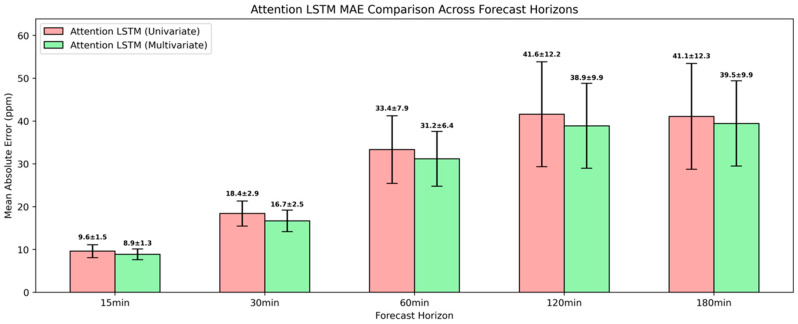
Evaluation of multivariate attention-based LSTM model in short-term CO_2_ forecasting using MAE and RMSE.

**Figure 17 sensors-25-05382-f017:**
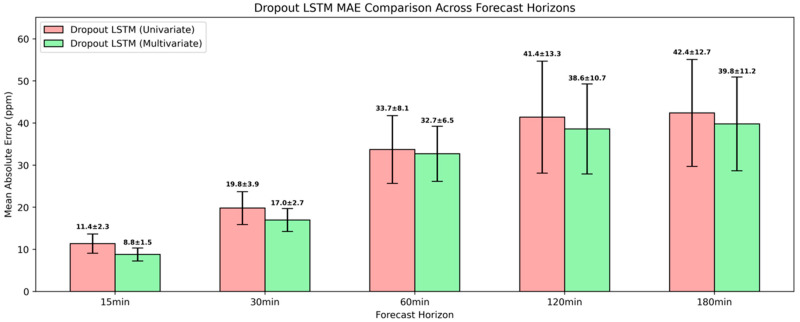
Evaluation of multivariate LSTM model with dropout in short-term CO_2_ forecasting using MAE and RMSE.

**Table 1 sensors-25-05382-t001:** The variables included in this study’s dataset.

Category	Variable	Notation	Units
Air pollutants	CO_2_	$y_{t}$	ppm
	PM_2.5_	$x_{t,\;pm}$	ug/m^3^
	TVOC	$x_{t,\;tv}$	ppb
Climatic conditions	Air temperature	$x_{t,\;tm}$	℃
	Relative humidity	$x_{t,\;rh}$	%
	Barometric pressure	$x_{t,\;bp}$	kPa
Occupancy status	Noise level	$x_{t,\;nl}$	dBA
	Luminosity	$x_{t,\;lm}$	lux

## Data Availability

The data presented in this study are available on request from the corresponding author due to privacy restrictions.

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
