# Peer review of "Short-Term Forecast of Indoor CO_2_ Using Attention-Based LSTM: A Use Case of a Hospital in Greece"

_sensors, 2025, doi:10.3390/s25175382_

Round 1

Reviewer 1 Report

Comments and Suggestions for Authors

Author Response

Reviewer 1

Comment 1: The introduction is quite lengthy; a more concise description of the section may be preferable.

Response: Thank you for the suggestion. The length of “Introduction” section has been shortened to provide a more concise description.

Comment 2: Combining the introduction and related work into a single section may boost understanding.

Response: Thank you for the suggestion. The “Introduction” and “Related Works” sections have been merged into a single section to enhance readability and understanding.

Comment 3: The method section requires a subsection that describes the system design, and including a flowchart or graphic that depicts the used design might considerably improve comprehension.

Response: Thank you for your remark. A graphical representation and a short description were added regarding the paper’s methodological approach.

Comment 4: The findings should be compared with the previous studies

Response: Thank you for your remark. In “Results and Discussion” section, a dedicated paragraph has been added, comparing our findings with results from prior studies.

Comment 5: The reference details in the reference list are incomplete.

Response: Thank you for your remark. The references list has been updated and now is compliant with the MDPI requirements for references.

Comment 6: Title Page: Lines 16 and 24: Take out the “hyphen”

Response: The text has been updated accordingly.

Comment 7: Pages 1-4: Ensure that the citations are accurately formatted.

Response: Citations have been formatted according to MDPI guidelines, e.g., [1], [1-3].

Comment 8: Pages 1-2: Lines 39-42: The USEPA stated that “EPA studies of human exposure to air pollutants indicate that indoor levels of pollutants may be two to five times — and occasionally more than 100 times — higher than outdoor levels”, so it’s not often 100 times higher.

Response: Thanks for the comment. The text has been revised to clarify that indoor air pollution is not often 100 times higher than outdoor levels.

Comment 9: Page 2: Lines 58-63: Add citation

Response: The relevant citation has been added.

Comment 10: Page 4: Lines 175-181: Remove the paragraph that doesn't contribute to the section.

Response: Paragraph has been removed.

Comment 11: Page 13: Line 528: Change the heading to Results and Discussion.

Response: Heading has been updated accordingly.

Comment 12 Page 13: Lines 529-537: Remove the paragraph that doesn't contribute to the section.

Response: Paragraph has been removed.

Reviewer 2 Report

Comments and Suggestions for Authors

The manuscript presents a well-structured study on the short-term prediction of indoor CO2 concentration using LSTM architectures with attention mechanisms, applied to a hospital setting. The combination of univariate and multivariate models, along with a comparative assessment of architectural complexities, provides useful insights into indoor air quality prediction.

The article is well aligned with the Sensors field and addresses a relevant and timely topic, especially considering the growing importance of indoor environmental monitoring in sensitive spaces such as hospitals.

The methodology and experimental setup are sound. It could be improved by making more concise and focusing the discussion more clearly on limitations and practical implications.

Minor Revisions

Does it make sense to use two decimal places for temperature values? Isn't one enough?
ppm values have uncertainties on the order of ones or tens, so it seems pointless to report even one decimal place, except for small values.

Some sentences overemphasize the results:
Example: "The results demonstrated that the LSTM with attention mechanism is the most robust predictive model for this particular task..."
 It would be more appropriate to write "Among the tested models, the attention-based LSTM exhibited the most consistent performance..." to avoid generalizing the "absolute superiority."
Stating "consistent superior performance" without mentioning limitations:
Some sentences emphasize the improvement over the baseline models, but fail to highlight that the dataset is specific to a single use case (a hospital).
 Include a caveat such as "within the scope of the studied environment" or "based on the specific dataset characteristics."
There is no discussion of possible biases arising from the sensor configuration and monitoring period:
The methodological structure is correct, but the text assumes that the model is easily generalizable.
 Explicitly state that further validation on external datasets is necessary.

Author Response

Comment 1: Does it make sense to use two decimal places for temperature values? Isn't one enough? ppm values have uncertainties on the order of ones or tens, so it seems pointless to report even one decimal place, except for small values.

Response: Thank you for your comment. Temperature values have been rounded to one decimal place. The same adjustment has been applied to TVOC values and to the reported uncertainties for predicted CO₂ concentrations.

Comment 2: Some sentences overemphasize the results: Example: "The results demonstrated that the LSTM with attention mechanism is the most robust predictive model for this particular task..." It would be more appropriate to write "Among the tested models, the attention-based LSTM exhibited the most consistent performance..." to avoid generalizing the "absolute superiority."

Response: Thank you for your comment. Throughout the manuscript, and in particular in the “Results and Discussion” section, sentences that overemphasized the results have been revised to avoid claims of absolute superiority, with the emphasis limited to comparisons among the examined LSTM models.

Comment 3: Stating "consistent superior performance" without mentioning limitations: Some sentences emphasize the improvement over the baseline models, but fail to highlight that the dataset is specific to a single use case (a hospital). Include a caveat such as "within the scope of the studied environment" or "based on the specific dataset characteristics."

Response: Thank you for your suggestion. The text has been updated (e.g., lines 655 and 665) to include appropriate caveats, clarifying that the conclusions are specific to the studied environment and dataset, and with emphasis restricted to the examined LSTM models.

Comment 4: There is no discussion of possible biases arising from the sensor configuration and monitoring period: The methodological structure is correct, but the text assumes that the model is easily generalizable. Explicitly state that further validation on external datasets is necessary.

Response: Thank you for your observation. The text has been revised (lines 698–703) to acknowledge possible biases from the sensor configuration and monitoring period, and to explicitly state the need for further validation on external datasets.

Comment 5: Explicitly state that further validation on external datasets is necessary.

Response: Thank you for your comment. The text has been updated (lines 701–703) to explicitly state this requirement.

Reviewer 3 Report

Comments and Suggestions for Authors

The manuscript by  Mountzouris et al. describes a study to forecast the CO2 concentration in a hospital. The statistics work seems to be reasonable and well documented. However, I am more concerned with the scientific values of this study. In particular:

1) Indoor air quality is affected by many factors other than the measured 8 type of data. The data including CO2 are the results not the causes. 

2) Trying to correlate the results to make prediction or forcast is not impossible, but the lack of causes just makes the study more statistical than science.

3) As the authors mentioned the occupany, ventalation, filtration, size of the space, air conditioning, and CO2 outside the zone are all very important, but difficult to include in the model.

4) It is unclear what the values of the forecast? Knowledge-based real time adjustment of the ventalation seems to be more important practically. Comparison to other similar forecast and applications needs to be mentioned.

5) In my opinions, the more important study would be find the major physical causes for the rise and decline of the CO2 concentration in the zone, and identify the dominant factors. Find a most economical way to keep the CO2 and PM2.5, VOC within health range.

Author Response

Point 1: Indoor air quality is affected by many factors other than the measured 8 type of data. The data including CO2 are the results not the causes.

Response: Thank you for your remark. We acknowledge that indoor air quality is influenced by many factors beyond the eight variables measured in this study. While CO₂ itself is an outcome rather than a direct cause, its strong inherent autocorrelation makes it a valuable predictor for short-term forecasting. In the context of a data-driven forecasting framework, including CO₂ among the input variables can significantly improve predictive accuracy for future CO₂ concentrations.

Point 2: Trying to correlate the results to make prediction or forecast is not impossible, but the lack of causes just makes the study more statistical than science.

Response: Thank you for your observation. We acknowledge that identifying the physical causes underlying CO₂ concentration variations would strengthen the scientific interpretation of the results. The present study was designed with a primary focus on developing and evaluating data-driven forecasting models, which inherently rely on statistical and machine learning approaches. While causal analysis was beyond the scope of this work, we agree it would complement the forecasting framework and enhance the interpretability of the predictions. To reflect this, we have added a note in the “Conclusions” section acknowledging the importance of integrating causal factor analysis in future research.

Point 3: As the authors mentioned the occupany, ventalation, filtration, size of the space, air conditioning, and CO2 outside the zone are all very important, but difficult to include in the model.

Response: Thank you for your comment. We agree that factors such as occupancy, ventilation, filtration, space size, air conditioning, and outdoor CO₂ levels are important yet challenging to capture directly. In this study, some of these factors were indirectly represented—for example, noise and light levels as proxies for occupancy, and other pollutant levels as indicators for ventilation and filtration. This indirect incorporation has been explicitly acknowledged as a limitation in the manuscript, and we note that direct measurement of these variables warrants further investigation. Within the scope of a data-driven ML approach, such proxy-based representation is considered sufficient for the present analysis.

Point 4: It is unclear what the values of the forecast? Knowledge-based real time adjustment of the ventilation seems to be more important practically. Comparison to other similar forecast and applications needs to be mentioned.

Response: Thank you for your remark. In “Results and Discussion” section, a dedicated paragraph has been added, comparing our findings with results from relevant studies and application for indoor CO2 forecasting.

Point 4: In my opinion, the more important study would be to find the major physical causes for the rise and decline of the CO2 concentration in the zone, and identify the dominant factors. Find a most economical way to keep the CO2 and PM2.5, VOC within health range.

Response: Thank you for your valuable suggestion. We agree that identifying the major physical causes of CO₂ concentration variations, as well as determining the dominant factors influencing PM₂.₅ and VOC levels, would provide important insights for effective environmental management. However, the primary objective of this study was to develop and evaluate forecasting models for indoor CO₂ concentrations, rather than to conduct a causal analysis. While a comprehensive investigation of physical causes and cost-effective control strategies lies beyond the present scope, we have included a statement in the “Conclusions” section highlighting this as a promising direction for future work.

Round 2

Reviewer 3 Report

Comments and Suggestions for Authors

The authors have made considerable effort in this revision, although the tracking information makes the manuscript really hard to read. I did not find the "a dedicated paragraph has been added, comparing our findings with results from relevant studies and application for indoor CO2 forecasting" in the results and discussion  section? Overall speaking, the manuscript is publishable if the authors can further provide the potential practical values of the models and the prediction for the control and improvement of the air quality for various indoor environment.

Author Response

Comment 1: The authors have made considerable effort in this revision, although the tracking information makes the manuscript really hard to read.

Response: We appreciate your comment. The tracked changes were automatically generated by the MDPI submission system and were not manually inserted by us. We acknowledge that this may make the manuscript harder to read; however, this process is system-driven and beyond our control.

Comment 2: I did not find the "a dedicated paragraph has been added, comparing our findings with results from relevant studies and application for indoor CO2 forecasting" in the results and discussion section?

Response: We apologize for the confusion, which may have been caused by the tracked changes in the revised manuscript. The dedicated comparison paragraph is indeed included between lines 763–775, where we discuss our findings in relation to relevant studies and applications for indoor CO₂ forecasting.

Comment 3: Overall speaking, the manuscript is publishable if the authors can further provide the potential practical values of the models and the prediction for the control and improvement of the air quality for various indoor environment.

Response: Thank you for your comment. We have updated the manuscript to include a dedicated paragraph outlining the potential practical values of the proposed models and their role in supporting prediction-based control and improvement of indoor air quality (lines 777–792).
